# Accurate bundle matching and generation via multitask learning with partially shared parameters

**Hyunsik Jeon, Jun-Gi Jang, Taehun Kim, U. Kang** *

Seoul National University, Seoul, Republic of Korea

* ukang@snu.ac.kr

## Abstract

How can we recommend existing bundles to users accurately? How can we generate new tailored bundles for users? Recommending a bundle, or a group of various items, has attracted widespread attention in e-commerce owing to the increased satisfaction of both users and providers. Bundle matching and bundle generation are two representative tasks in bundle recommendation. The bundle matching task is to correctly match existing bundles to users while the bundle generation is to generate new bundles that users would prefer. Although many recent works have developed bundle recommendation models, they fail to achieve high accuracy since they do not handle heterogeneous data effectively and do not learn a method for customized bundle generation. In this paper, we propose BUNDLEMAGE, an accurate approach for bundle matching and generation. BUNDLEMAGE effectively mixes user preferences of items and bundles using an adaptive gate technique to achieve high accuracy for the bundle matching. BUNDLEMAGE also generates a personalized bundle by learning a generation module that exploits a user preference and the characteristic of a given incomplete bundle to be completed. BUNDLEMAGE further improves its performance using multi-task learning with partially shared parameters. Through extensive experiments, we show that BUNDLEMAGE achieves up to 6.6% higher nDCG in bundle matching and 6.3× higher nDCG in bundle generation than the best competitors. We also provide qualitative analysis that BUNDLEMAGE effectively generates bundles considering both the tastes of users and the characteristics of target bundles.

## Introduction

*Given item and bundle purchase histories of users, how can we match existing bundles to the users and generate new bundles for them?* Recommending a bundle, or a group of various items, instead of individual items has attracted widespread attention in e-commerce since 1) it recommends items that users would prefer at once and 2) it increases the chances of unpopular items being exposed to users. Bundle recommendation is divided into two different but highly related tasks, bundle matching and bundle generation, both of which play important roles in bundle recommendation. Bundle matching, which is to accurately match pre-constructed

**Data Availability Statement:** The data underlying this study are available on GitHub (https://github.com/snudatalab/BundleMage).

**Funding:** This work was supported by Jung-Hun Foundation. The Institute of Engineering Research

at Seoul National University provided research facilities for this work. The ICT at Seoul National University provides research facilities for this study. The funders had no role in study design, data collection and analysis, decision to publish, or preparation of the manuscript."

bundles to users, is crucial because it reduces the cost of manually constructing a bundle every time. Bundle generation, which automatically generates personalized bundles for users, is necessary because it enables us to construct a new bundle that better reflects user preferences than the pre-constructed bundles in a long-term perspective.

Bundle recommendation, however, is challenging due to the following reasons. First, bundle matching requires careful handling of heterogeneous types of data (i.e., user-item interactions and user-bundle interactions) to extract meaningful preferences of users. Previous works [1–5] fail to achieve high accuracy for bundle matching since they do not establish a relationship between the heterogeneous data. Second, bundle generation is a demanding task since the search space of possible bundles is burdensome to cope with; finding all possible bundles requires exponential computational costs to the number of items. Existing methods [1, 5] do not learn any generation mechanism from the observable data. Instead, they heuristically generate new personalized bundles based on a learned bundle matching model and show poor performance on bundle generation as a result. Third, it requires careful design of architecture to achieve high accuracy in both bundle matching and generation since they are highly related but different tasks. Previous works [1–5] have not studied architectures that perform both tasks concurrently since they have focused only on the bundle matching model.

In this paper, we propose BUNDLEMAGE (Accurate Bundle Matching and Generation via Multitask Learning with Partially Shared Parameters), an accurate method for bundle recommendation. To achieve high accuracy for the bundle matching, BUNDLEMAGE carefully aggregates information of user-bundle and user-item interactions by exploiting an adaptive gate technique which adaptively balances the contribution of heterogeneous information. BUNDLEMAGE also learns a generation mechanism to provide a new tailored bundle for users. We train a generation module of BUNDLEMAGE by reconstructing given incomplete bundles, exploiting the preferences of users who have interacted with them. BUNDLEMAGE further improves its performance via multi-task learning with partially shared parameters to address the bundle matching and bundle generation problems simultaneously. With these ideas, BUNDLEMAGE accurately recommends existing bundles to users, and successfully generates new bundles that users would prefer.

Our contributions are summarized as follows:

- **Method**. We propose BUNDLEMAGE, an accurate method for personalized bundle matching and generation. BUNDLEMAGE accurately matches users to bundle using their past item and bundle interactions. BUNDLEMAGE also effectively generates personalized bundles using target users' preferences.

- **Experiments**. Extensive experiments on real-world datasets show that BUNDLEMAGE provides the state-of-the-art performance with up to 6.6% higher nDCG in bundle matching, and up to 6.3× higher nDCG in bundle generation compared to the best competitors (see Tables 3 and 4).

- **Case studies**. We show in case studies that BUNDLEMAGE successfully generates personalized bundles even with unpopular items which would otherwise be rarely exposed (see Figs 1 and 7).

The code and datasets are available at https://github.com/snudatalab/BundleMage. Symbols used frequently in this paper are summarized in Table 1.

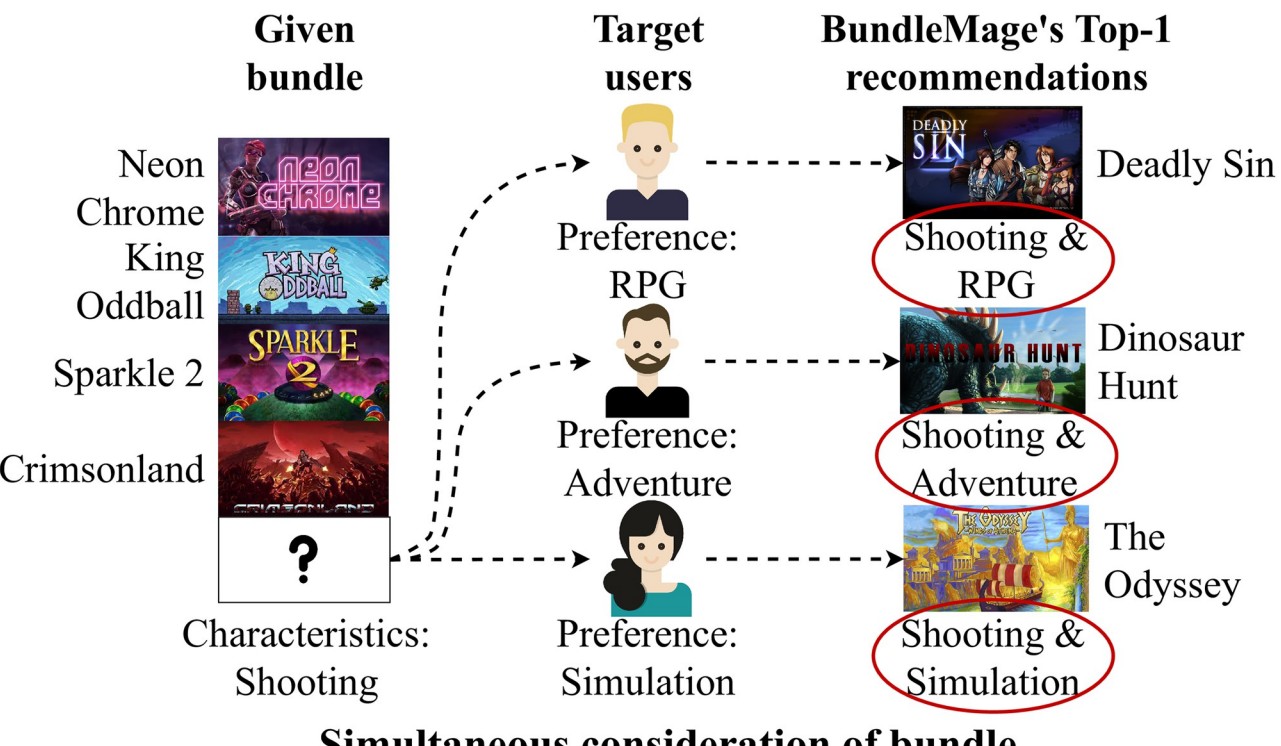

**Fig 1. Top-1 recommendations of BundleMage for different target users in bundle generation.** BUNDLEMAGE considers the characteristics of a given bundle and the preferences of target users for bundle generation. For instance, BUNDLEMAGE recommends a shooting and RPG game (e.g., Deadly Sin) for a bundle of shooting games and a target *user A* who prefers RPG.

## Problem definition

Bundle recommendation [6] aims to predict bundles, instead of items, that a user would prefer. For each user $u$, we observe item interaction vector $\mathbf{v}_u \in \mathbb{R}^{N_i}$ and bundle interaction vector $\mathbf{r}_u \in \mathbb{R}^{N_b}$, where $N_i$ and $N_b$ are the numbers of items and bundles, respectively. $\mathbf{v}_u$ and $\mathbf{r}_u$ are binary vectors, where each nonzero entry indicates the interaction with the corresponding item or bundle. We have a binary bundle-item affiliation matrix $\mathbf{X} \in \mathbb{R}^{N_i \times N_b}$ where each non-zero entry indicates the inclusion of an item to a bundle; $\mathbf{x}_b \in \mathbb{R}^{N_i}$, which indicates $b$th column of $\mathbf{X}$, is the item affiliation vector of bundle $b$. We denote the sets of indices of observable entries in $\mathbf{v}_u$, $\mathbf{r}_u$, and $\mathbf{x}_b$ as $\Omega(\mathbf{v}_u) = \{i : i \in \mathcal{I}\}$, $\Omega(\mathbf{r}_u) = \{b : b \in \mathcal{B}\}$, and $\Omega(\mathbf{x}_b) = \{i : i \in \mathcal{I}\}$, respectively; $\mathcal{U}, \mathcal{I}$, and $\mathcal{B}$ are the sets of users, items, and bundles, respectively. We describe the formal definition of bundle matching and bundle generation as follows.

*Problem* 1 (Bundle matching):

**Given**: a user $u$'s item interaction vector $\mathbf{v}_u$ and bundle interaction vector $\mathbf{r}_u$,

**Predict**: the user $u$'s next interacted bundle $b'$, where $b' \in \mathcal{B}$ and $b' \notin \Omega(\mathbf{r}_u)$.

*Problem* 2 (Bundle generation):

**Given**: a user $u$'s item interaction vector $\mathbf{v}_u$, bundle interaction vector $\mathbf{r}_u$, and an incomplete bundle $\tilde{\mathcal{G}} = \{i : i \in \mathcal{I}\}$ to be completed,

**Construct**: a personalized bundle $\mathbb{G}(u, \tilde{\mathcal{G}}) = \{i' : i' \in \mathcal{I}, i' \notin \tilde{\mathcal{G}}\}$ of size $k \ll |\mathcal{I}|$, which denotes a small set of items to complete $\tilde{\mathcal{G}}$ for user $u$, to be recommended to user $u$ as the complete set $\tilde{\mathcal{G}} \cup \mathbb{G}(u, \tilde{\mathcal{G}})$.

**Bundle matching module** (Section III-B)

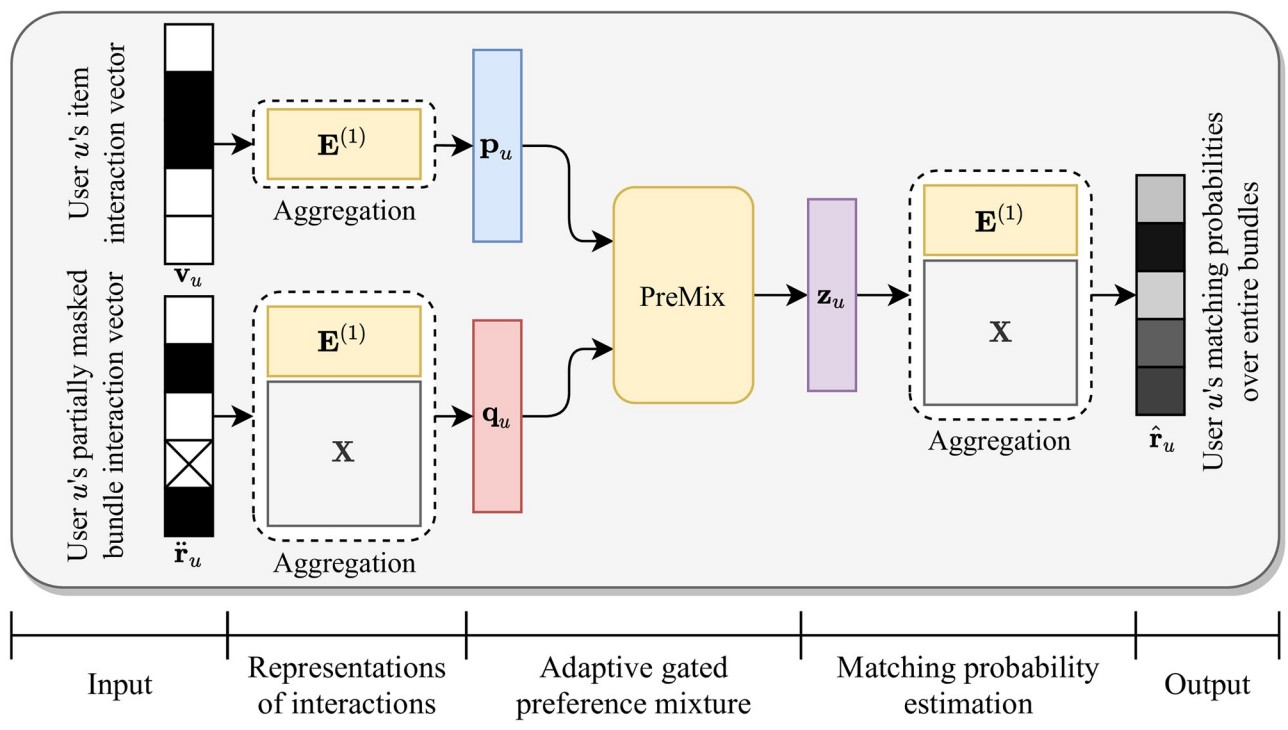

**Fig 2. The architecture of bundle matching module in BundleMage.**

**Bundle generation module** (Section III-C)

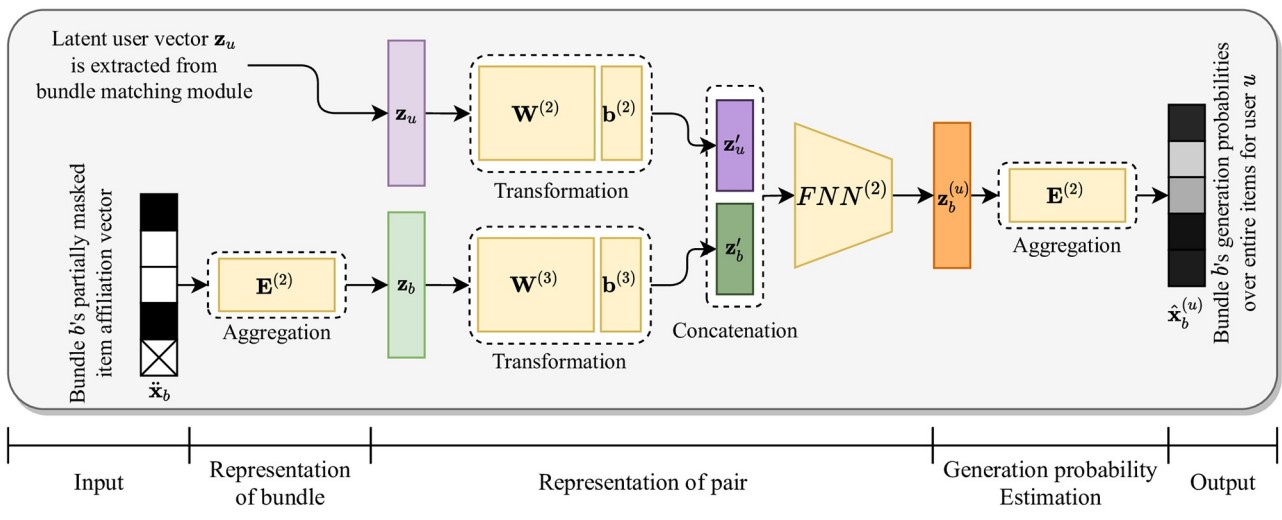

**Fig 3. The architecture of bundle generation module in BundleMage.**

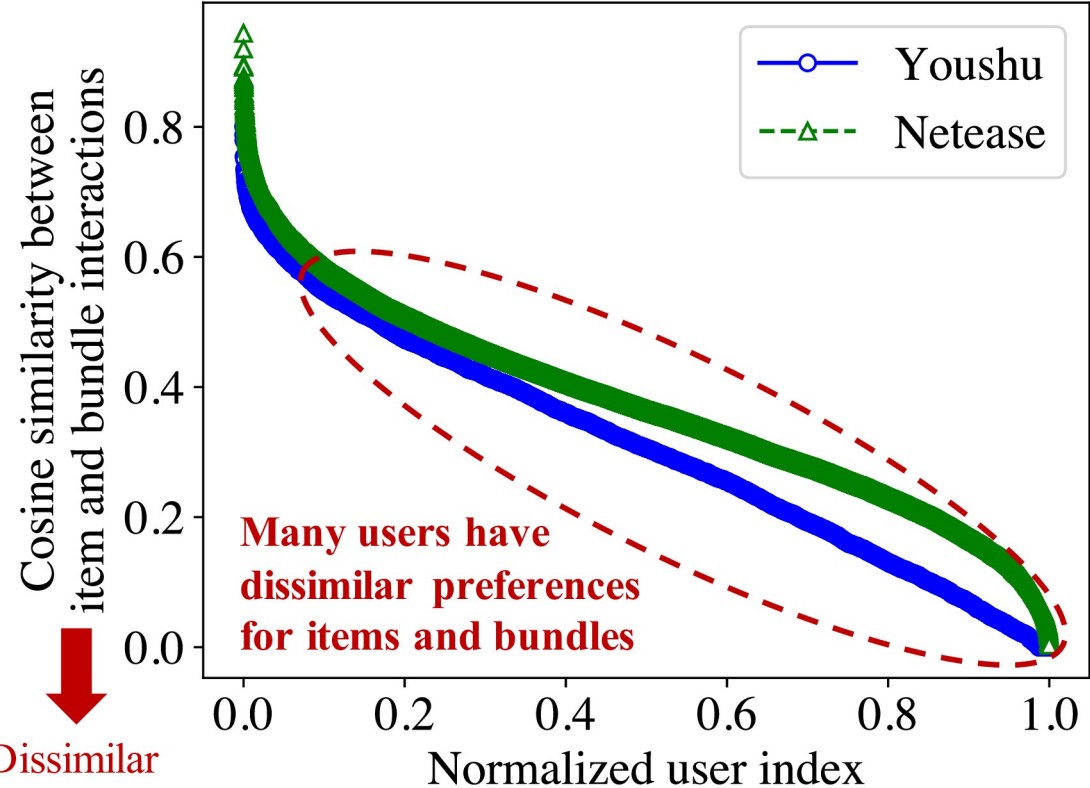

**Fig 4. Cosine similarities between item and bundle interactions of each user in two real-world datasets.** A lot of users have dissimilar preferences for items and bundles.

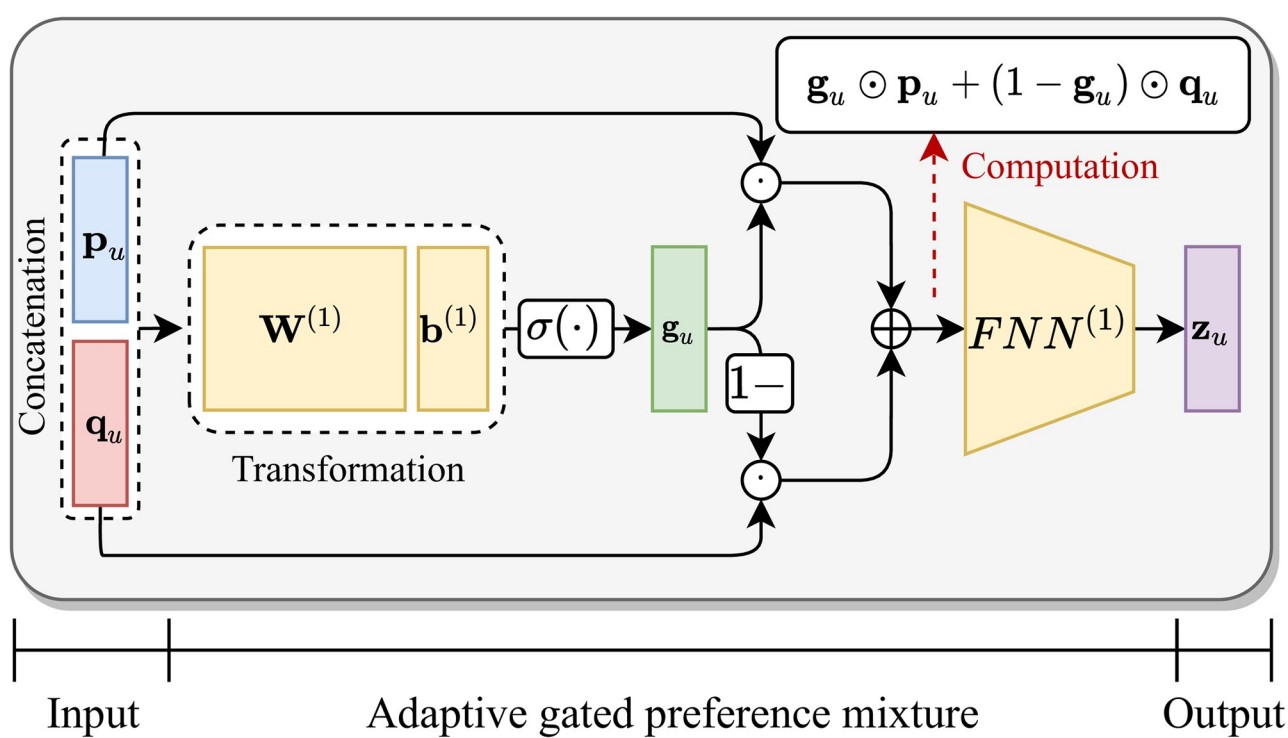

**Fig 5. The architecture of adaptive gated preference mixture (PreMix).**

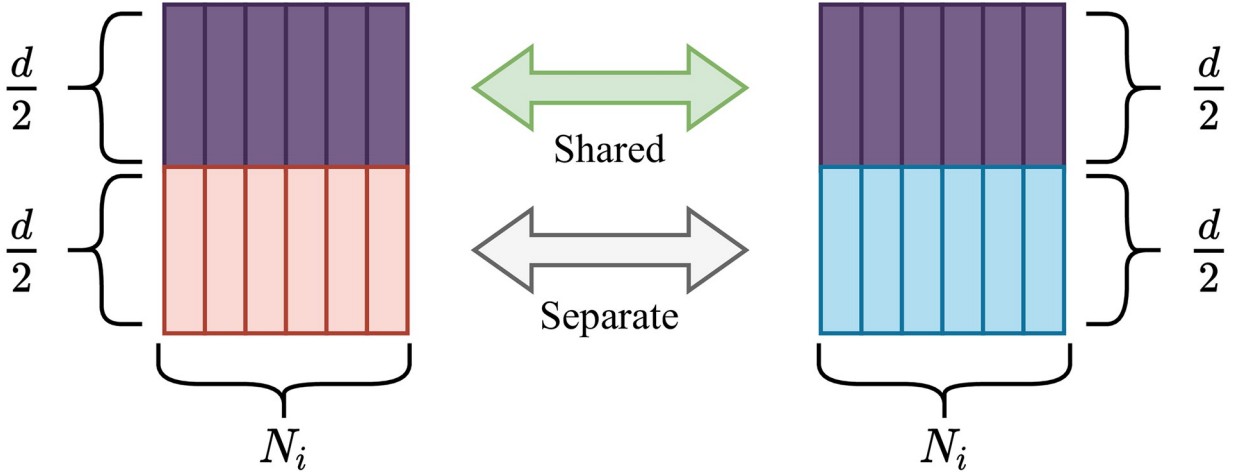

**Fig 6. An illustration of shared parameters of E$^{(1)}$ and E$^{(2)}$, which are item embedding vectors of bundle matching and generation modules, respectively.**

## Related works

In this section, we summarize related works of this work.

### Collaborative filtering

Collaborative filtering is the most extensively used recommendation approach due to its powerful performance in real world services. Collaborative filtering predicts items a user would

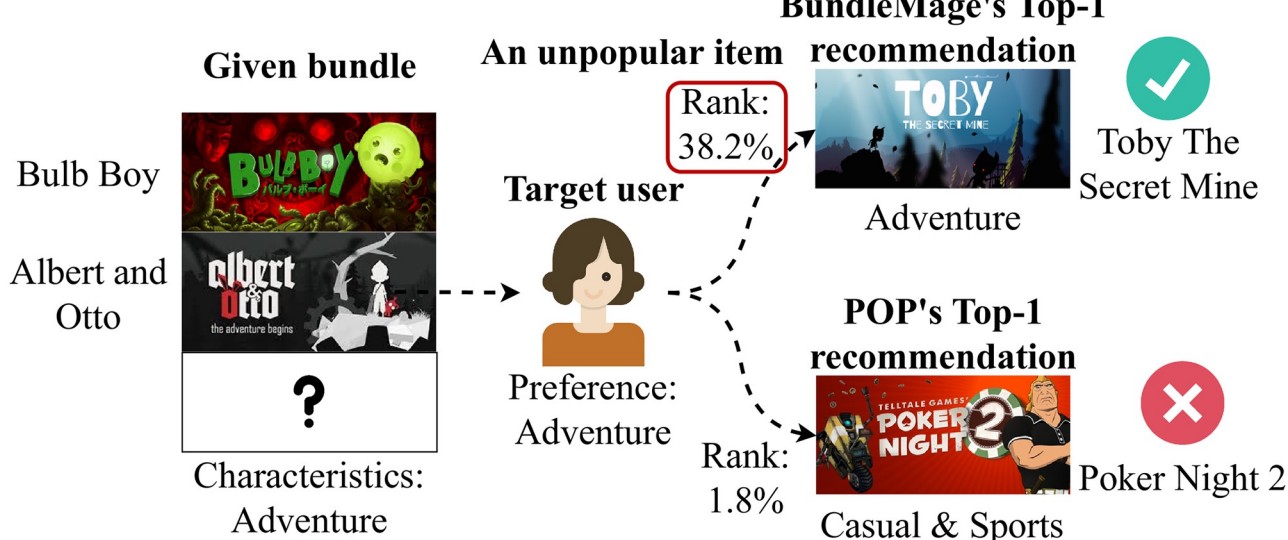

**Fig 7. Top-1 recommendations of BundleMage and POP for a target user in bundle generation.** BUNDLEMAGE successfully recommends an unpopular item "Toby: The Secret Mine" (ranked in top 38.2%) which is the ground-truth one, whereas POP recommends a popular item "Poker Night 2" (ranked in top 1.8%) which is unrelated to the given bundle and the target user.

**Table 1. Table of frequently used symbols.**

| Symbol | Description |
|:---:|:---|
| $\mathbf{v}_u$ | user $u$'s item interaction vector ($\in N_i$) |
| $\mathbf{r}_u$ | user $u$'s bundle interaction vector ($\in N_b$) |
| $\ddot{\mathbf{r}}_u$ | user $u$'s partially masked bundle interaction vector ($\in N_b$) |
| $\mathbf{x}_b$ | bundle $b$'s item affiliation vector ($\in N_i$) |
| $\ddot{\mathbf{x}}_b$ | bundle $b$'s partially masked item affiliation vector ($\in N_i$) |
| $N_u, N_i, N_b$ | numbers of users, items, and bundles, respectively |
| $\Omega(\mathbf{v}_u), \Omega(\mathbf{r}_u), \Omega(\mathbf{x}_b)$ | indices of observable entries in $\mathbf{v}_u$, $\mathbf{r}_u$, and $\mathbf{x}_b$, respectively |
| $\mathcal{U}, \mathcal{I}, \mathcal{B}$ | sets of users, items, and bundles, respectively |

prefer by capturing similar patterns across users and items. On early works, matrix factorization approaches [7–9] learn latent factors of users and items while predicting interactions by a linear way. They still largely prevail recommender system community because of their simplicity and effectiveness. Recent collaborative filtering-based approaches utilize deep neural networks to embody the non-linear properties of users' interactions. NCF [10] learns a non-linear scoring function as well as latent factors using fully-connected neural networks. AutoRec [11] learns an autoencoder to learn latent representations of users' interactions. CDAE [12] adopts a denoising autoencoder [13] to improve robustness of top-$N$ recommendation performance. VAE-CF [14] extends a variational autoencoder [15] to collaborative filtering to learn meaningful manifold of user preferences. However, the item collaborative filtering methods are not entirely suitable for bundle recommendation since they have not handled bundles which are more challenging to deal with than individual items.

## Bundle recommender systems

For the bundle matching task, early works have adopted BPR framework [9] to learn latent factors of users, items, and bundles by optimizing a pairwise ranking loss. BR [1] learns latent factors of users and items from user-item interactions using the BPR loss, and predicts users' bundle interactions by aggregating the latent item factors. EFM [2] jointly factorizes user-item and user-bundle interaction matrices using the BPR loss; it further incorporates item-item co-occurrence information to improve the performance. DAM [3] adopts an attention mechanism to represent latent bundle factors, extends NCF structure [10] to a multi-task learning framework, and learns user-item and user-bundle interactions using the BPR loss. Recent works have leveraged Graph Convolutional Networks [16] to learn user-item-bundle relationships from a unified heterogeneous graph. BGCN [4, 17] constructs a heterogeneous graph consisting of user, item, and bundle nodes, and learns latent factors of the nodes while propagating the information of interactions and affiliations. GRAM-SMOT [5] adopts Graph Attention Networks [18] to reflect the relative influence of items in a bundle. However, such bundle matching methods have not considered that users may have different interaction patterns for items and bundles. For instance, a user may purchase an item if it is included in a bundle even if she would not have purchased it individually. Thus, we expect performance improvement when considering the heterogeneous preference for items and bundles. For the bundle generation task, BR [1] and GRAM-SMOT [5] have tried to generate personalized bundles for users. However, they bypass the problem by generating items in a greedy manner through trained bundle matching models instead of learning a generation mechanism from observable data. In addition, there has been no study for a unified architecture of bundle matching and

generation; if matching and generation tasks are trained together, performance improvement is expected for both tasks since they are different but highly related tasks.

## Proposed method

In this section, we propose BUNDLEMAGE (Accurate <u>Bundle</u> <u>Mat</u>ching and <u>Ge</u>neration via Multitask Learning with Partially Shared Parameters), an accurate method for personalized bundle recommendation.

### Overview

We address the following challenges to achieve a high performance of the bundle recommendation.

C1. **Handling heterogeneous interactions**. Users have heterogeneous interactions with items and bundles, both of which are informative but dissimilar. How can we effectively extract user preferences from the heterogeneous interactions for accurate bundle matching?

C2. **Learning customized bundle generation**. Bundle generation is a demanding task since the search space of possible bundles is prohibitively unwieldy. Moreover, personalized bundle generation is necessary since each user has a different taste for bundles. How can we generate bundles customized for a target user?

C3. **Handling two related but different tasks**. Bundle matching and generation are related but separate tasks. How can we effectively train a model to improve the performance of the two tasks simultaneously?

The main ideas of BUNDLEMAGE are summarized as follows:

I1. **(Bundle Matching) Adaptive gated preference mixture** in a bundle matching module enables us to effectively represent user preferences from heterogeneous interactions with items and bundles.

I2. **(Bundle Generation) Learning the reconstruction** of an incomplete bundle using user preference enables us to generate personalized bundles.

I3. **Multi-task learning with partially shared parameters** enables us to learn the common and separate information of matching and generation tasks, and results in high performance on the two tasks simultaneously.

BUNDLEMAGE consists of bundle matching module and bundle generation module. Figs 2 and 3 depict the architectures of bundle matching and bundle generation modules, respectively. As shown in Fig 2, the bundle matching module is trained to predict a user's entire bundle interactions using the user's entire item interactions and a part of bundle interactions. In the module, an adaptive gated preference mixture (PreMix) adaptively integrates the user's heterogeneous interactions for items and bundles. It effectively exploits user preferences for bundle matching from the heterogeneous interactions. As shown in Fig 3, the bundle generation module is trained to complete a bundle's affiliations from incomplete ones, using the latent factor of a user who has interacted with the bundle. The generation module is able to learn a personalized bundle generation mechanism since it is trained to reconstruct bundles for a user under observed interaction pairs of users and bundles. The bundle matching and generation modules are trained in a multi-task learning manner while sharing parts of item

embedding vectors. It successfully improves the performance of matching and generation simultaneously.

## Bundle matching

The objective of bundle matching is to predict bundles a user would prefer using her past item and bundle interactions. For the bundle matching, it is important to effectively extract users' preferences from the item and bundle interactions. However, users may differently interact with items and bundles since items and bundles are inherently different. Before describing our method for bundle matching, we investigate the interaction patterns from real-world datasets to verify users have dissimilar preferences for items and bundles. Specifically, we compute cosine similarities between users' item and bundle interactions to measure how consistent users' preferences are for items and bundles. Fig 4 shows cosine similarities between item and bundle interactions of each user in two real-world datasets, Youshu and Netease (details in Table 2). We compute the cosine similarities by the following procedure. First, for each item $i$, we obtain a user interaction multi-hot vector $\mathbf{c}_i \in \mathbb{R}^{N_u}$ where $N_u$ is the number of users; each nonzero entry in $\mathbf{c}_i$ indicates the interaction of the corresponding user. Note that items with similar vectors are more likely to be similar to each other since it means they have many over-lapping interacted users. Second, for each user $u$, we compute an item preference vector as $\frac{1}{|\Omega(\mathbf{v}_u)|} \sum_{i \in \Omega(\mathbf{v}_u)} \mathbf{c}_i$, where $\mathbf{v}_u \in \mathbb{R}^{N_i}$ is user $u$'s item interaction vector, $N_i$ is the number of items, and $\Omega(\mathbf{v}_u)$ is the set of indices of nonzero entries in $\mathbf{v}_u$. Also, for each user $u$, we compute a bundle preference vector as $\frac{1}{|\mathcal{S}_u|} \sum_{i \in \mathcal{S}_u} \mathbf{c}_i$, where $\mathcal{S}_u = \cup_{b \in \Omega(\mathbf{r}_u)} \Omega(\mathbf{x}_b)$, $\mathbf{r}_u \in \mathbb{R}^{N_b}$ is user $u$'s bundle interaction vector, $N_b$ is the number of bundles, $\mathbf{x}_b \in \mathbb{R}^{N_i}$ is bundle $b$'s item affiliation vector, and $\Omega(\mathbf{r}_u)$ and $\Omega(\mathbf{x}_b)$ are the sets of indices of nonzero entries in $\mathbf{r}_u$ and $\mathbf{x}_b$, respectively. Note that each user's preference is computed as the averaged vector of items or bundles that the user interacted with to represent the user's preferences for items and bundles. Last, we compute cosine similarity between the item and bundle preference vectors of each user, and sort them in descending order. As shown in Fig 4, a plenty of users have dissimilar interaction patterns for items and bundles. In order to accurately match bundles to users, we design a matching module while considering that users have different preferences for items and bundles.

The main challenge of bundle matching is to extract meaningful user preference from het-erogeneous interactions for items and bundles, which entail dissimilar patterns. Meanwhile, both of interactions are crucial for predicting bundles that a user would prefer, because they both represent the preference of the user. Thus, the main technical difficulty in bundle match-ing is integrating the heterogeneous interactions to represent user preferences and accurately matching bundles to them. Our main idea is to adaptively balance the information of two interactions. Fig 2 depicts the structure of bundle matching module. The matching module 1) represents a user's item and bundle interactions as low-dimensional latent factors, 2) integrates

**Table 2. Summary of datasets.**

| Dataset | Users | Bundles | Items | User-bundle (dens.) | User-item (dens.) | Bundle-item (dens.) | Avg. size of bundles |
|---|---|---|---|---|---|---|---|
| Youshu[1] | 8,039 | 4,771 | 32,770 | 51,377 (0.13%) | 138,515 (0.05%) | 176,667 (0.11%) | 37.03 |
| Netease[2] | 18,528 | 22,864 | 123,628 | 302,303 (0.07%) | 1,128,065 (0.05%) | 1,778,838 (0.06%) | 77.80 |
| Steam[3] | 29,634 | 615 | 2,819 | 87,565 (0.48%) | 902,967 (1.08%) | 3,541 (0.20%) | 5.76 |

[1] https://github.com/yliuSYSU/DAM

[2] https://github.com/cjx0525/BGCN

[3] https://github.com/technoapurva/Steam-Bundle-Recommendation

the latent factors using an adaptive gated preference mixture (PreMix), and 3) estimates matching probabilities over bundles.

**Representations of interactions.** For each user $u$, we have item interaction vector $\mathbf{v}_u \in \mathbb{R}^{N_i}$ and bundle interaction vector $\mathbf{r}_u \in \mathbb{R}^{N_b}$, where $N_i$ and $N_b$ are the numbers of items and bundles, respectively. Note that $\mathbf{v}_u$ and $\mathbf{r}_u$ are multi-hot binary vectors, where each nonzero entry indicates the interaction with the corresponding item or bundle. We obtain the representation vector $\mathbf{p}_u \in \mathbb{R}^d$ of user $u$'s item interactions by the average of item embeddings that $u$ has interacted with:

$$\mathbf{p}_u = \frac{1}{\|\mathbf{v}_u\|_1} \mathbf{E}^{(1)} \mathbf{v}_u, \tag{1}$$

where $\mathbf{E}^{(1)} \in \mathbb{R}^{d \times N_i}$ is a trainable item embedding matrix for bundle matching; each column in $\mathbf{E}^{(1)}$ indicates the embedding vector of the corresponding item. We also obtain the representation vector $\mathbf{q}_u \in \mathbb{R}^d$ of user $u$'s bundle interactions by the average of embedding vectors of bundles that user $u$ has interacted with. To obtain it, we use a partially masked vector $\ddot{\mathbf{r}}_u \in \mathbb{R}^{N_b}$, which we obtain from $\mathbf{r}_u$ by masking $0 \leq \rho \leq 1$ ratio of nonzero entries to zeros. This enables the matching module to learn to predict the masked nonzero entries as well as the unobserved nonzero entries, which is advantageous for accurately recommending new bundles to users at test. The representation vector $\mathbf{q}_u \in \mathbb{R}^d$ is obtained as follows:

$$\mathbf{q}_u = \frac{1}{\|\ddot{\mathbf{r}}_u\|_1} \mathbf{E}^{(1)} \mathbf{X} \mathbf{D}^{-1} \ddot{\mathbf{r}}_u, \tag{2}$$

where $\mathbf{X} \in \mathbb{R}^{N_i \times N_b}$ is a bundle-item affiliation matrix and $\mathbf{D} \in \mathbb{R}^{N_b \times N_b}$ is a diagonal matrix whose $i$th diagonal element $\mathbf{D}_{ii}$ is equal to $\sum_j \mathbf{X}_{ji}$. $\mathbf{E}^{(1)} \mathbf{X} \mathbf{D}^{-1} \in \mathbb{R}^{d \times N_b}$ indicates the embedding matrix of bundles where each column represents the embedding vector of the corresponding bundle since each bundle embedding is computed as the average of embeddings of its constituent items. As a result, Eq (2) is the average of embeddings of the observed bundles in $\ddot{\mathbf{r}}_u$.

**Adaptive gated preference mixture.** The main challenge is to mix the two representation vectors $\mathbf{p}_u$ and $\mathbf{q}_u$ well, to accurately predict the next bundle interactions of a user $u$. To address the challenge, we propose an adaptive gated preference mixture (PreMix) which adaptively balances the two vectors. As illustrated in Fig 5, PreMix integrates the two representation vectors while adaptively balancing their contributions, and obtains a latent user vector $\mathbf{z}_u \in \mathbb{R}^d$ as follows:

$$
\begin{aligned}
\mathbf{g}_u &= \sigma\left(\mathbf{W}^{(1)} \begin{bmatrix} \mathbf{p}_u \\ \mathbf{q}_u \end{bmatrix} + \mathbf{b}^{(1)}\right), \\
\mathbf{z}_u &= \mathrm{FNN}^{(1)}(\mathbf{g}_u \odot \mathbf{p}_u + (1 - \mathbf{g}_u) \odot \mathbf{q}_u),
\end{aligned}
\tag{3}
$$

where $\mathbf{g}_u \in \mathbb{R}^d$ is a gate vector, $\sigma(\cdot)$ is the sigmoid function, $\mathbf{W}^{(1)} \in \mathbb{R}^{d \times 2d}$ and $\mathbf{b}^{(1)} \in \mathbb{R}^d$ are a trainable weight matrix and a bias vector, respectively, and the square bracket $[\cdot]$ denotes concatenation. $\mathrm{FNN}^{(1)}(\cdot)$ is a 2-layered feed forward neural network with the structure $\mathbb{R}^d \to \mathbb{R}^{\frac{d}{2}} \to \mathbb{R}^d$ containing an activation function, and $\odot$ indicates the element-wise product. We employ the neural network to extract more complicated non-linear features from the gated preference mixture. Furthermore, we constrain the hidden dimension of the neural network to be smaller than the input and output dimensions to effectively extract meaningful information from the input. A high value of $\mathbf{g}_u$ indicates that the information of item interactions has a great influence on matching the next bundle to the user $u$.

**Matching probability estimation.**    For evaluation, we need to obtain predicted matching probabilities $\hat{\mathbf{r}}_u \in \mathbb{R}^{N_b}$ for a user $u$. We first compute matching scores for every bundle using the latent user vector $\mathbf{z}_u$ and embedding vectors of bundles $\mathbf{E}^{(1)}\mathbf{X}\,\mathbf{D}^{-1}$. Then, the predicted matching probabilities $\hat{\mathbf{r}}_u$ of a user $u$ are obtained by normalizing the scores with the softmax function:

$$\hat{\mathbf{r}}_u = \text{softmax}\!\left((\mathbf{E}^{(1)}\mathbf{X}\mathbf{D}^{-1})^{\top}\mathbf{z}_u\right) \tag{4}$$

where softmax($\cdot$) is the softmax function and $\mathbf{E}^{(1)}\mathbf{X}\mathbf{D}^{-1} \in \mathbb{R}^{d\times N_b}$ is the embedding matrix of bundles.

## Bundle generation

The objective of bundle generation is to construct a personalized bundle for a target user. Bundle generation is a demanding task since the search space of possible bundles is prohibitively unwieldy. Previous works for bundle generation [1, 5] detour the problem by utilizing pretrained bundle matching models in a greedy manner without training any bundle generation mechanism. However, they have limitation of necessitating a heuristic criterion for whether to add new items or remove existing items from a bundle being generated. To address such limitation, it is necessary to train a personalized bundle generation model from the observable user-bundle interactions. Our main idea is to train a model to reconstruct a bundle from an incomplete one, using a target user's preference. Our intuition is that a bundle construction is determined by the characteristic of the bundle and the preference of a target user; the characteristics of bundles could vary by domain, such as genre or provider. For instance, assume we want to generate a bundle with PlayStation games, and the target user prefers RPG (Roll Playing Games). We then need to construct a bundle with PlayStation and RPG games for the target user. Fig 3 depicts the structure of bundle generation module. Given a pair of a user and her interacted bundle, the generation module 1) represents the bundle's incomplete affiliations as low-dimensional latent factors, 2) obtains a hidden representation of the pair, and 3) estimates the incomplete bundle's generation probabilities over items for the target user.

**Representation of bundle.**    Given user $u$ and her interacted bundles $b \in \Omega(\mathbf{r}_u)$, our idea for bundle generation is to train a model to reconstruct bundle $b$'s original affiliations from an incomplete ones using $u$'s preference. For each bundle $b$, we have item affiliation vector $\mathbf{x}_b \in \mathbb{R}^{N_i}$, where $N_i$ is the number of items; $\mathbf{x}_b$ is $b$th column of bundle-item affiliation matrix $\mathbf{X}$. To represent bundle $b$'s incomplete affiliations, we define $\ddot{\mathbf{x}}_b \in \mathbb{R}^{N_i}$ where we mask $0 \le \psi \le 1$ ratio of nonzero entries in $\mathbf{x}_b$ to zeros. We start with obtaining a low-dimensional representation vector $\mathbf{z}_b \in \mathbb{R}^d$ of the incomplete bundle affiliation vector $\ddot{\mathbf{x}}_b$ by the average of item embeddings:

$$\mathbf{z}_b = \frac{1}{\|\ddot{\mathbf{x}}_b\|_1}\mathbf{E}^{(2)}\ddot{\mathbf{x}}_b, \tag{5}$$

where $\mathbf{E}^{(2)} \in \mathbb{R}^{d\times N_i}$ is a trainable item embedding matrix for bundle generation; analogous to $\mathbf{E}^{(1)}$, each column in $\mathbf{E}^{(2)}$ indicates the embedding vector of the corresponding item. The masking strategy enables the generation module to learn to predict the masked nonzero entries as well as the unobserved nonzero entries, which is advantageous for accurately generating new items at test.

**Representation of pair.**    To predict items that are appropriate for the given bundle $b$ and user $u$, we need to represent the pair of bundle $b$ and user $u$ as a vector by exploiting their information. Given bundle $b$'s representation vector $\mathbf{z}_b$ and user $u$'s representation vector $\mathbf{z}_u$,

we obtain the representation vector $\mathbf{z}_b^{(u)} \in \mathbb{R}^d$ of a pair of user $u$ and bundle $b$ by performing linear transformation independently, and integrating them using a feed forward neural network. Note that we use the latent user vector $\mathbf{z}_u$ extracted from Eq (3) since it represents user $u$'s preference. The detail is described as follows:

$$\mathbf{z}'_u = \mathbf{W}^{(2)}\mathbf{z}_u + \mathbf{b}^{(2)}, \qquad \mathbf{z}'_b = \mathbf{W}^{(3)}\mathbf{z}_b + \mathbf{b}^{(3)},$$
$$\mathbf{z}_b^{(u)} = \text{FNN}^{(2)}\left( \begin{bmatrix} \mathbf{z}_{u'} \\ \mathbf{z}_{b'} \end{bmatrix} \right), \tag{6}$$

where $\mathbf{z}'_u, \mathbf{z}'_b \in \mathbb{R}^d$ are linearly transformed vectors from $\mathbf{z}_u$ and $\mathbf{z}_b$, respectively, $\mathbf{W}^{(2)} \in \mathbb{R}^{\frac{d}{2} \times d}$ and $\mathbf{W}^{(3)} \in \mathbb{R}^{\frac{d}{2} \times d}$ are trainable weight matrices, $\mathbf{b}^{(2)} \in \mathbb{R}^{\frac{d}{2}}$ and $\mathbf{b}^{(3)} \in \mathbb{R}^{\frac{d}{2}}$ are trainable bias vectors, and $\text{FNN}^{(2)}(\cdot)$ is a 2-layered feed forward neural network with the structure $\mathbb{R}^d \to \mathbb{R}^{\frac{d}{2}} \to \mathbb{R}^d$.

**Generation probability estimation.** We estimate the generation probability distribution over items for the pair of user $u$ and bundle $b$ as follows:

$$\hat{\mathbf{x}}_b^{(u)} = \text{softmax}(\mathbf{E}^{(2)\top}\mathbf{z}_b^{(u)}), \tag{7}$$

where $\hat{\mathbf{x}}_b^{(u)} \in \mathbb{R}^d$ is the predicted bundle generation probability over items, for user $u$ given an incomplete bundle $b$. $\mathbf{E}^{(2)}$ is the embedding matrix of items which is used also in representing $\mathbf{z}_b$.

## Multi-task learning with partially shared parameters

Our goal is to maximize the performance of the two tasks, bundle matching and generation. The two tasks are different but highly related, which inevitably entail common information as well as separate information. Thus, the main technical difficulty is effectively learning the shared and separate information for the two tasks. Our main idea is to train the bundle matching and generation module in a multi-task learning manner while sharing parts of parameters.

**Partially shared parameters.** Parameter sharing technique is broadly studied and incentivized in multi-task learning since it bestows an advantage of impressive performance [19, 20]. However, imprudent sharing rather decreases the performance of two different tasks [21, 22]. As shown in Fig 6, we thus propose to share parts of item embedding vectors to achieve high performance on bundle matching and generation simultaneously. We denote $i$'th column of $\mathbf{E}^{(1)}$ and $\mathbf{E}^{(2)}$ as $\mathbf{e}_i^{(1)} \in \mathbb{R}^d$ and $\mathbf{e}_i^{(2)} \in \mathbb{R}^d$, respectively; they represent item $i$'s embedding vectors for bundle matching and generation, respectively. We share halves of $\mathbf{e}_i^{(1)}$ and $\mathbf{e}_i^{(2)}$ with the same parameters while letting the other halves trained separately. This enables BundleMage to learn the common and separate information for the two tasks, resulting in improving the performance of the two tasks simultaneously. We conduct thorough experiments on parameter sharing to show that our method is effective in improving the performance of bundle matching and generation, which is described in the following section.

**Objective function for multi-task learning.** Our goal is to obtain optimal parameters $\mathbf{E}^{(1)}, \mathbf{E}^{(2)}, \mathbf{W}^{(1)}, \mathbf{W}^{(2)}, \mathbf{W}^{(3)}, \mathbf{b}^{(1)}, \mathbf{b}^{(2)}, \mathbf{b}^{(3)}, \text{FNN}^{(1)}$, and $\text{FNN}^{(2)}$ to accurately estimate the matching and generation probabilities. Thus, we optimize the parameters to minimize the distance between the predicted probability and the ground-truth probability. For the bundle matching and generation tasks, we utilize multinomial likelihoods for distributions $\mathbf{r}_u$ and $\mathbf{x}_b$ as in previous works [14, 23–28], since it has shown more impressive results than other likelihoods such as Gaussian likelihood and logistic likelihood in top-$k$ recommendation [14]. Thus, the losses are measured by KL-divergence between the observed probabilities and the predicted

probabilities. Specifically, the loss to be minimized for bundle matching is defined as follows:

$$\mathcal{L}_{mat} = -\frac{1}{N_u} \sum_{u \in \mathcal{U}} \frac{1}{\|\mathbf{r}_u\|_1} \sum_{b \in \Omega(\mathbf{r}_u)} \mathbf{r}_{ub} \log \hat{\mathbf{r}}_{ub}, \tag{8}$$

where $\mathcal{L}_{mat}$ is the bundle matching loss, $\mathbf{r}_{ub}, \hat{\mathbf{r}}_{ub} \in \mathbb{R}$ are $b$th elements in $\mathbf{r}_u \in \mathbb{R}^{N_b}$ and $\hat{\mathbf{r}}_u \in \mathbb{R}^{N_b}$, respectively. Analogously, the loss to be minimized for bundle generation is defined as follows:

$$\mathcal{L}_{gen} = -\frac{1}{N_u} \sum_{u \in \mathcal{U}} \frac{1}{\|\mathbf{r}_u\|_1} \sum_{b \in \Omega(\mathbf{r}_u)} \frac{1}{\|\mathbf{x}_b\|_1} \sum_{i \in \Omega(\mathbf{x}_b)} \mathbf{x}_{bi} \log \hat{\mathbf{x}}_{bi}, \tag{9}$$

where $\mathcal{L}_{gen}$ is the bundle generation loss, $\mathbf{x}_{bi}, \hat{\mathbf{x}}_{bi} \in \mathbb{R}$ are $i$th elements in $\mathbf{x}_b \in \mathbb{R}^{N_i}$ and $\hat{\mathbf{x}}_b \in \mathbb{R}^{N_i}$, respectively. To minimize the bundle matching loss and the bundle generation loss simultaneously, we define the objective function to be minimized as follows:

$$\mathcal{L}_{rec} = \mathcal{L}_{mat} + \mathcal{L}_{gen}, \tag{10}$$

where $\mathcal{L}_{rec}$ is the objective function. Note that the matching and generation modules are trained to reconstruct the entire nonzero entries in $\mathbf{r}_u$ and $\mathbf{x}_b$, respectively, although they use the masked vector $\ddot{\mathbf{r}}_u$ and $\ddot{\mathbf{x}}_b$, respectively, as inputs. It makes the modules to accurately predict the unobserved interactions and affiliations. In practice, we iteratively minimize the bundle matching loss $\mathcal{L}_{mat}$ and the bundle generation loss $\mathcal{L}_{gen}$ in every epoch.

## Experiments

In this section, we perform experiments to answer the following questions.

Q1. **Bundle matching**. Does BUNDLEMAGE show higher accuracy in bundle matching than those of baselines?

Q2. **Bundle generation**. Does BUNDLEMAGE generate a personalized bundle for a target user well?

Q3. **Ablation study**. How the modules in BUNDLEMAGE help improve the performance of BUNDLEMAGE?

Q4. **Case study**. What bundles does BUNDLEMAGE generate for users?

### Experimental setup

We introduce our experimental setup including datasets, baseline approaches, evaluation metrics, the training process, and hyperparameters.

**Datasets.**   We use three real-world bundle recommendation datasets as summarized in Table 2. Youshu [3] contains bundles (sets of books) from a book review site. Netease [2] contains bundles (sets of musics) from a cloud music service. Steam [1] contains bundles (sets of video games) from a video game distribution platform.

**Baselines.**   We compare BUNDLEMAGE with existing methods for the two tasks: bundle matching and bundle generation. There are nine existing methods for bundle matching as follows.

- **POP** recommends the top-$k$ popular bundles to users.

- **BPR** [9] is a matrix factorization method under a Bayesian Personalized Ranking learning framework.

- **NCF** [10] is a neural network-based model which combines a generalized matrix factorization and neural networks to capture the high-order interactions between users and bundles.

- **VAE-CF** [14] extends Variational Autoencoder [15] to collaborative filtering and maximizes the multinomial likelihood of user interactions.

- **BR** [1] learns the latent vectors of users and items under Bayesian Personalized Ranking and learns the latent vectors of bundles aggregating the learned latent item vectors in a linear way.

- **EFM** [2] jointly factorizes the user-item-bundle interaction matrix and item-item-bundle co-occurrence information matrix.

- **DAM** [3] uses the attention mechanism and multi-task learning framework to learn users', items', and bundles' latent vectors.

- **BGCN** [4] unifies user-item interactions, user-bundle interactions, and bundle-item affiliations into a heterogeneous graph and trains a Graph Convolutional Network [16] on it to predict affinities between users and bundles.

- **GRAM-SMOT** [5] also constructs a heterogeneous graph and trains a Graph Attention Network [29] by a metric learning approach [30].

We also compare BUNDLEMAGE with the following four existing methods for the bundle generation task.

- **Random** randomly chooses $k$ items.

- **POP** chooses the top-$k$ popular items.

- **BR** [1] repeatedly adds the best item to an incomplete bundle by computing the user-bundle score with a trained bundle matching model.

- **GRAM-SMOT** [5] picks items close to a target user greedily; closeness is measured by latent vectors of the target user and items.

Note that BR and GRAM-SMOT work based on their learned bundle matching modules. We use only a user-bundle interaction matrix for BPR, NCF, and VAE-CF due to their modeling capability. On the other hand, we use all given matrices for BR, EFM, DAM, BGCN, and GRAM-SMOT.

**Evaluation metrics.** We evaluate the performance of bundle matching and bundle generation with two evaluation metrics, recall@$k$ and normalized discounted cumulative gain (nDCG@$k$), which are the most widely used metrics for evaluating accuracy as in previous works [4, 31]. For each user, both metrics compare the predicted rank of the held-out items with their true rank. While recall@$k$ considers all items ranked within the first $k$ to be equally important, nDCG@$k$ considers higher ranks more importantly by monotonically increasing the discount factor. We vary $k$ in {5, 10, 20} for all datasets.

**Experimental process.** To evaluate the generation performance for unseen bundles, we randomly select 10% of bundles. We use user-bundle interactions of the selected bundles as the test held-out for bundle generation to evaluate the generation performance on bundles that have not been observed in training. For the rest of the user-bundle interactions, we employ *leave-one-out* protocol [3, 5, 10, 32–35] to split them into training, matching validation, and matching test datasets. Specifically, we randomly select two bundles for each user, and one is used as a matching validation held-out and the other is used as a matching test held-out. For bundle matching task, we randomly select 99 bundles that have not been

interacted with each user as negative samples to compare with the validation and test bundles following previous works [3, 5, 10]. For bundle generation task, we randomly select $n$ items from each bundle as positive samples in the generation test held-out. We also randomly select $m$ items that is not contained in each bundle as negative samples to compare with the positive samples. We set $(n, m)$ to (1, 99), (5, 495), and (10, 990) for Steam, Youshu, and Netease datasets, respectively. We report experimental results for the matching and generation test held-outs when a model shows the best nDCG@5 on the matching validation dataset within 200 epochs. We run each experiment at least three times and report the average.

**Hyperparameters.** We set the masking ratios $\rho$ and $\psi$ to 0.5, We set the learning rate to 0.001 among {0.01, 0.001, 0.0001, 0.0001}, the weight decay to 0.00001 among {0.001, 0.0001, 0.00001, 0.000001}, and dropout rate [36] to 0.3 among {0.1, 0.3, 0.5, 0.7, 0.9}. Note that the hyperparameters are set to the best one among the candidates. We set embedding dimensionality $d$ of all methods to 200 for a fair comparison. We use Adam optimizer [37] for the training.

## Performance on bundle matching (Q1)

We evaluate the performance of BundleMage and competitors for the bundle matching. Table 3 shows the results in terms of Recall@$k$ and nDCG@$k$. We have two main observations. First, BundleMage shows the best performance in most cases, achieving up to 6.6% higher nDCG than the competitors. Second, our modeling of how we handle heterogeneous types of interaction data is more effective on a large dataset than on a small dataset; note that Bundle-Mage adaptively extracts user preferences from the heterogeneous interactions with items and bundles. The performance gap is large between BundleMage and the competitors for Netease and Steam datasets which have plenty of user-item and user-bundle interactions. BundleMage effectively extracts user preferences from those interactions for the bundle matching. In contrast, the performance gap is not large on Youshu dataset because it contains less user-item and user-bundle interactions compared to the other datasets.

**Table 3. Performance of BundleMage and competitors for bundle matching with respect to nDCG and Recall.**

| Model | nDCG@$k$ | | | | | | | | | Recall@$k$ | | | | | | | | |
| | Youshu | | | Netease | | | Steam | | | Youshu | | | Netease | | | Steam | | |
| | @5 | @10 | @20 | @5 | @10 | @20 | @5 | @10 | @20 | @5 | @10 | @20 | @5 | @10 | @20 | @5 | @10 | @20 |
|---|---|---|---|---|---|---|---|---|---|---|---|---|---|---|---|---|---|---|
| POP | .0264 | .0451 | .0681 | .0315 | .0479 | .0723 | .0207 | .0531 | .0596 | .0490 | .1062 | .1982 | .0528 | .1040 | .2017 | .0407 | .1012 | .1312 |
| BPR [9] | .3828 | .4196 | .4439 | .2796 | .3255 | .3647 | .8407 | .8450 | .8462 | .5172 | .6306 | .7260 | .3979 | .5402 | .6952 | .9791 | .9921 | .9969 |
| NCF [10] | .4651 | .5047 | .5269 | .3142 | .3605 | .3976 | .8497 | .8535 | .8545 | .6152 | .7365 | .8240 | .4448 | .5878 | .7347 | .9830 | .9942 | .9984 |
| VAE-CF [14] | .4788 | .5150 | .5394 | .4026 | .4466 | .4786 | .9182 | .9212 | .9219 | .6323 | .7439 | .8402 | <u>.5571</u> | .6919 | .8139 | .9874 | <u>.9965</u> | <u>.9990</u> |
| BR [1] | .4319 | .4710 | .4969 | .2764 | .3248 | .3659 | .8425 | .8465 | .8476 | .5853 | .7058 | .8075 | .3977 | .5476 | .7103 | .9824 | .9946 | .9988 |
| EFM [2] | .4541 | .4941 | .5189 | .2918 | .3406 | .3814 | .8472 | .8513 | .8522 | .6058 | .7289 | .8263 | .4162 | .5676 | .7289 | .9823 | .9948 | .9984 |
| DAM [3] | .4520 | .4937 | .5203 | .2893 | .3362 | .3772 | .8509 | .8549 | .8564 | .6052 | .7328 | .8374 | .4117 | .5568 | .7198 | .9815 | .9939 | **.9995** |
| BGCN [4] | .3811 | .4231 | .4512 | .3256 | .3692 | .4062 | .9001 | .9035 | .9043 | .5420 | .6713 | .7821 | .4513 | .5861 | .7324 | .9851 | .9952 | .9983 |
| GRAM-SMOT [5] | <u>.5018</u> | <u>.5405</u> | <u>.5628</u> | <u>.4051</u> | <u>.4501</u> | <u>.4823</u> | <u>.9225</u> | <u>.9253</u> | <u>.9259</u> | <u>.6568</u> | <u>.7750</u> | **.8627** | <u>.5571</u> | <u>.6961</u> | <u>.8230</u> | <u>.9878</u> | .9963 | .9985 |
| **BundleMage (proposed)** | **.5185** | **.5533** | **.5740** | **.4281** | **.4724** | **.5039** | **.9838** | **.9849** | **.9852** | **.6693** | **.7770** | <u>.8584</u> | **.5762** | **.7132** | **.8373** | **.9947** | **.9978** | <u>.9990</u> |

BundleMage outperforms all competitors in most cases, demonstrating its superiority of personalized bundle matching. Bold and underlined values indicate the best and the second best accuracies, respectively.

**Table 4. Performance of BundleMage and competitors for bundle generation with respect to nDCG and Recall.**

| Model | nDCG@$k$ | | | | | | | | | Recall@$k$ | | | | | | | | |
|---|---|---|---|---|---|---|---|---|---|---|---|---|---|---|---|---|---|---|
| | Youshu | | | Netease | | | Steam | | | Youshu | | | Netease | | | Steam | | |
| | @5 | @10 | @20 | @5 | @10 | @20 | @5 | @10 | @20 | @5 | @10 | @20 | @5 | @10 | @20 | @5 | @10 | @20 |
| Random | .0078 | .0135 | .0221 | .0100 | .0100 | .0155 | .0266 | .0425 | .0669 | .0081 | .0185 | .0390 | .0051 | .0101 | .0201 | .0453 | .0953 | .1930 |
| POP | .2078 | .2525 | .3179 | .1266 | .1110 | .1430 | .7698 | .7765 | .7770 | .1935 | .2843 | .4354 | .0623 | .1031 | .1616 | **.9673** | **.9871** | **.9891** |
| BR [1] | .0082 | .0134 | .0251 | .0104 | .0104 | .0159 | .0322 | .0481 | .0729 | .0080 | .0180 | .0460 | .0051 | .0103 | .0203 | .0549 | .1044 | .2037 |
| GRAM-SMOT [5] | .0114 | .0175 | .0272 | o.o.t | o.o.t | o.o.t | .0963 | .1222 | .1412 | .0029 | .0083 | .0120 | o.o.t | o.o.t | o.o.t | .0733 | .1556 | .2734 |
| **BundleMage (proposed)** | **.4592** | **.5634** | **.6187** | **.7944** | **.6974** | **.7842** | **.9650** | **.9657** | **.9671** | **.4018** | **.5942** | **.7217** | **.3891** | **.6448** | **.7993** | .9660 | .9682 | .9738 |

BUNDLEMAGE outperforms all competitors in most cases, demonstrating its superiority of personalized bundle generation. Bold and underlined values indicate the best and the second best accuracies, respectively. We mark experiments that take more than 7 days as o.o.t (out of time).

## Performance on bundle generation (Q2)

We evaluate the performance of the bundle generation in terms of Recall@$k$ and nDCG@$k$. In Table 4, BUNDLEMAGE provides the state-of-the-art accuracy by achieving up to 6.3× higher nDCG than the competitors. Note that the performance gap is large since only BUNDLEMAGE learns a generation mechanism for personalized bundles from the observable data. To show the popularity biases of datasets, we measure the average of every bundle's ranking score which is evaluated as the average of rankings of included items. For each dataset, the averaged scores are measured as follows: Youshu (14.8%), Netease (26.36%), and Steam (1.42%). For the bundle generation, POP has a good performance than Random, BR, and GRAM-SMOT since many bundles consist of popular items. The performance of POP is especially good for Steam dataset because of its extreme popularity bias. However, BUNDLEMAGE outperforms POP in most cases, since BUNDLEMAGE accurately generates bundles consisting of unpopular items as well as popular ones.

## Ablation study (Q3)

For an ablation study, we compare the accuracy of BUNDLEMAGE and its variants to evaluate whether each module in BUNDLEMAGE helps the performance improvement. The variants of BUNDLEMAGE are as follows.

- **BundleMage-Avg**. To evaluate the effect of the adaptive gated preference mixture module, we incorporate representation vectors as $\frac{1}{2}(\mathbf{p}_u + \mathbf{q}_u)$ instead of $\mathbf{g}_u \odot \mathbf{p}_u + (1 - \mathbf{g}_u) \odot \mathbf{q}_u$ in Eq (3).

- **BundleMage-Sep**. To evaluate the partially shared parameters technique, we entirely separate the parameters of $\mathbf{E}^{(1)}$ and $\mathbf{E}^{(2)}$.

- **BundleMage-Sha**. To evaluate the partially shared parameters technique, we entirely share the parameters of $\mathbf{E}^{(1)}$ and $\mathbf{E}^{(2)}$.

- **BundleMage-$\mathcal{L}_{gen}$**. To evaluate the multi-task learning technique, we train BUNDLEMAGE without $\mathcal{L}_{gen}$.

- **BundleMage-$\mathcal{L}_{mat}$**. To evaluate the multi-task learning technique, we train BUNDLEMAGE without $\mathcal{L}_{mat}$.

**Table 5. Evaluation of BundleMage and its variants for bundle matching with respect to nDCG.**

| Model | Youshu | | | Netease | | | Steam | | |
|---|---|---|---|---|---|---|---|---|---|
| | @5 | @10 | @20 | @5 | @10 | @20 | @5 | @10 | @20 |
| BundleMage-Avg | .5034 | .5404 | .5622 | .4219 | .4700 | .5012 | .9721 | .9737 | .9742 |
| BundleMage-Sep | .4880 | .5239 | .5457 | .4086 | .4558 | .4881 | .9612 | .9633 | .9638 |
| BundleMage-Sha | .3645 | .4041 | .4324 | .3359 | .3796 | .4149 | .9802 | .9812 | .9816 |
| BundleMage-$\mathcal{L}_{gen}$ | .4894 | .5283 | .5507 | .4108 | .4567 | .4896 | .9797 | .9811 | .9816 |
| **BundleMage** | **.5185** | **.5533** | **.5740** | **.4281** | **.4724** | **.5039** | **.9838** | **.9849** | **.9852** |

**Bundle matching.** For the bundle matching, we compare BundleMage with its variants BundleMage-Avg, BundleMage-Sep, BundleMage-Sha, and BundleMage-$\mathcal{L}_{gen}$. Table 5 shows the result of the ablation study for bundle matching. Note that BundleMage shows better performance than its variants, indicating that the adaptive gated preference mixture, partially shared parameters, and multi-task learning improve the performance of the bundle matching.

**Bundle generation.** For the bundle generation, we compare BundleMage with BundleMage-Avg, BundleMage-Sep, BundleMage-Sha, and BundleMage-$\mathcal{L}_{mat}$. Table 6 shows the result of the ablation study for bundle generation. As in the ablation study for bundle matching, using the adaptive gated mixture, multi-task learning, and partially shared parameters improves the performance of the bundle generation, Specifically, a latent user vector extracted from bundle matching modules plays an important role in generating bundles since removing the matching module from BundleMage degrades the performance of bundle generation.

## Case study (Q4)

We show in case studies that BundleMage successfully generates a personalized bundle even using unpopular items which would otherwise be rarely exposed. Fig 1 shows that BundleMage differently completes the bundle depending on target users when an incomplete bundle is given. Note that the incomplete bundle consists of shooting games. BundleMage adds the shooting and RPG game in the given bundle for *user A* interested in games of the RPG genre while adding the shooting and adventure game for *user B* interested in adventure genre games. For *user C* who prefers games of the simulation genre, BundleMage adds the shooting and simulation game in the given incomplete bundle. BundleMage successfully generates a new bundle by considering user preferences and characteristics of bundles.

We provide another case study of bundle generation for a comparison between BundleMage and POP. As shown in Fig 7, BundleMage correctly completes a bundle by considering the characteristics of the bundle while POP does not; in contrast to POP, BundleMage

**Table 6. Evaluation of BundleMage and its variants for bundle generation with respect to nDCG.**

| Model | Youshu | | | Netease | | | Steam | | |
|---|---|---|---|---|---|---|---|---|---|
| | @5 | @10 | @20 | @5 | @10 | @20 | @5 | @10 | @20 |
| BundleMage-Avg | .4328 | .5181 | .5815 | .7826 | .6911 | .7817 | .9642 | .9643 | .9648 |
| BundleMage-Sep | .4211 | .5100 | .5829 | .7831 | .6866 | .7743 | .9645 | .9648 | .9661 |
| BundleMage-Sha | .4304 | .4997 | .5686 | .7832 | .6904 | .7756 | .9643 | .9649 | .9666 |
| BundleMage-$\mathcal{L}_{mat}$ | .3534 | .4324 | .4922 | .7469 | .6521 | .7468 | .8199 | .8201 | .8202 |
| **BundleMage** | **.4592** | **.5634** | **.6187** | **.7944** | **.6974** | **.7842** | **.9650** | **.9657** | **.9671** |

successfully recommends an unpopular adventure game since the given bundle includes adventure games and the target user prefers adventure games.

## Conclusion

In this paper, we propose BUNDLEMAGE, an accurate model to simultaneously perform bundle matching and generation. BUNDLEMAGE matches bundles to users by effectively extracting users' preferences from their heterogeneous interactions with items and bundles. BUNDLEMAGE also generates a tailored bundle for a target user by exploiting a given incomplete bundle's characteristics and the preference of the target user. To further improve accuracy for the two tasks simultaneously, BUNDLEMAGE is trained in a multi-task learning manner with partially shared parameters. We experimentally show that BUNDLEMAGE achieves up to 6.6% higher nDCG in bundle matching and 6.3× higher nDCG in bundle generation than existing bundle recommendation models. Moreover, we experimentally verify that our main ideas of adaptive gated preference mixture, partially shared parameters, and multi-task learning improve the performance both of bundle matching and generation. Especially, we show that the matching module has a great influence on the generation performance, demonstrating the importance of the multi-task learning-based approach in the two related tasks, matching and generation. Our case studies show that BUNDLEMAGE 1) differently completes bundles depending on target users, and 2) generates personalized bundles even using un-popular items. Future works include extending BUNDLEMAGE to exploit auxiliary information of users, items, and bundles.

## Author Contributions

**Conceptualization:** Hyunsik Jeon.

**Data curation:** Hyunsik Jeon.

**Formal analysis:** Hyunsik Jeon, Jun-Gi Jang, U. Kang.

**Funding acquisition:** U. Kang.

**Investigation:** Hyunsik Jeon, Jun-Gi Jang, Taehun Kim.

**Methodology:** Hyunsik Jeon.

**Project administration:** U. Kang.

**Resources:** Hyunsik Jeon, U. Kang.

**Software:** Hyunsik Jeon.

**Supervision:** U. Kang.

**Validation:** Hyunsik Jeon, Jun-Gi Jang.

**Visualization:** Hyunsik Jeon.

**Writing – original draft:** Hyunsik Jeon.

**Writing – review & editing:** Hyunsik Jeon, Jun-Gi Jang, Taehun Kim, U. Kang.

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
