## [Decision Letter · Decision Letter 0]

25 Nov 2022

PONE-D-22-30342Accurate Bundle Matching and Generation via Multitask Learning with Partially Shared ParametersPLOS ONE

Dear Dr. Kang,

Thank you for submitting your manuscript to PLOS ONE. After careful consideration, we feel that it has merit but does not fully meet PLOS ONE’s publication criteria as it currently stands. Therefore, we invite you to submit a revised version of the manuscript that addresses the points raised during the review process.

We look forward to receiving your revised manuscript.

Kind regards,

Sathishkumar V E

Academic Editor

PLOS ONE

Journal Requirements:

"This work was supported by Jung-Hun Foundation. The Institute of Engineering Research at Seoul National University provided research facilities for this work. The ICT at Seoul National University provides research facilities for this study."

Reviewers' comments:

Reviewer's Responses to Questions

**Comments to the Author**

1. Is the manuscript technically sound, and do the data support the conclusions?

Reviewer #1: Partly

Reviewer #2: Yes

2. Has the statistical analysis been performed appropriately and rigorously? 

Reviewer #1: No

Reviewer #2: Yes

3. Have the authors made all data underlying the findings in their manuscript fully available?

Reviewer #1: Yes

Reviewer #2: Yes

4. Is the manuscript presented in an intelligible fashion and written in standard English?

Reviewer #1: No

Reviewer #2: Yes

5. Review Comments to the Author

Reviewer #1: In this submission the authors present two methodologies for bundle recommendation: one for bundle creation and another for bundle recommendation (matching). Their work is intuitive and original, especially w.r.t the PreMix module. However, a number of issues need to be addressed prior to accepting this manuscript for publication.

1. Please, organize your manuscript better. Avoid asking too many questions and use a more natural and scientific language flow. In this respect, I believe Figure 1(a) should be placed near the results section of the manuscript, while Figure 1(b) (the intuition for your work) should remain at its place. Additionally, problem definition should be a different section, preceded by the related works sections. Finally proofread your manuscript for the correct use of the english language and for correcting typos (eg “performance” instead of “performances” on line 23).

2. The analysis of lines 168-188 (different item & bundle preferences for each users) is not convincing, it it does not examine the intra-bundle cosine similarities of items.

3. Eq (2) should be explained in more detail

4. The adaptive gated preference mixture is among the top contribution of this manuscript, therefore it needs to be analyzed much more, instead of just laying out equations. For example what is the intuition behind the input of the FNN in Eq. 3. Why is this form the most suitable?

5. Even though a leave-one-out cross-validation seems to have been considered in the experiments (lines 334-348) w.r.t to bundle matching, it appears that this is not the case for bundle generation, since it is reported that a random 10% of the bundles has been singled out as a test set. Additionally, the statistical significance of the reported results on Tables 3-5 is not assessed.

6. The concluding section is too thin. It should at least report the main points and the “lessons learned” from the overall approach.

Reviewer #2: The manuscript is well written with supporting results and explanations

The literature has to be strongly updated with some relevant and recent papers focused on the fields dealt with in the manuscript and making related works as a separate section will be more appropriate for the readers.

Any hyperparamter search done? Authors set default hyperparaters suggested by the articles in reference. Provide explanation regarding this.

Why recall@k and nDCG@k is used as a performance metric?

6. PLOS authors have the option to publish the peer review history of their article (what does this mean?). If published, this will include your full peer review and any attached files.

Reviewer #1: No

Reviewer #2: **Yes: **Usha Moorthy

---

## [Author Response · Author response to Decision Letter 0]

3 Jan 2023

We would like to thank the reviewers for their high quality reviews and constructive comments. Below, we summarize how we addressed the reviewers’ requirements.

1. Reviewer 1.

• (R1-1) Please organize your manuscript better. Avoid asking too many questions and use a more natural and scientific language flow. In this respect, I believe Fig 1(a) should be placed near the results section of the manuscript, while Fig 1(b) (the intuition for your work) should remain at its place. Additionally, problem definition should be a different section, preceded by the related works sections. Finally proofread your manuscript for the correct use of the english language and for correcting typos (e.g., performance instead of performances on line 23).

– (A1-1) We replaced the interrogative sentences with other scientific sentences in the section on the proposed method (lines 195-197 and lines 262-263) except in the overview where we provide the research question of this work. We removed Fig 1(a) because it would be redundant if placed near the results section, while Fig 1(b) remained in its place as you recommended. We separated the sections of problem definition and related works in lines 56-120. We also reviewed the overall manuscript to correct typos.

• (R1-2) The analysis of lines 168-188 (different item & bundle preferences for each users) is not convincing, it does not examine the intra-bundle cosine similarities of items.

– (A1-2) Our goal of the analysis in the section on bundle matching is to compare user preferences for items and bundles. We extract the representative vector of each bundle as the average of the vectors of the constituent items regardless of the intra-bundle relationships of the items. For instance, if a bundle mainly consists of action games, the bundle’s vector is computed to represent the action games by averaging the vectors of the constituent items. Furthermore, each user’s preference is computed as the averaged vector of items or bundles that the user interacted with. Thus, our analysis provides dissimilar preference patterns of users for items and bundles by comparing the user interactions for items and bundles. We clearly provided the description in lines 184-186 of the bundle matching section.

• (R1-3) Equation (2) should be explained in more detail.

– (A1-3) Equation (2) denotes the average of embeddings of the observed bundles in $\\ddot{\\mathbf{r}}_b$. We provided a more detailed description of the equation in lines 212-214 of the bundle matching section.

• (R1-4) The adaptive gated preference mixture is among the top contribution of this manuscript, therefore it needs to be analyzed much more, instead of just laying out equations. For example, what is the intuition behind the input of the FNN in Equation (3)? Why is this form the most suitable?

– (A1-4) The intuition behind the usage of the FNN is to extract more complicated non-linear features from the gated preference mixture. We also constrain the hidden dimension of the neural network to be smaller than the input and output dimensions to effectively extract only meaningful information. We provided the intuition and the description of the FNN in lines 218-222 of the bundle matching section.

• (R1-5) Even though a leave-one-out cross-validation seems to have been considered in the experiments (lines 334-348) w.r.t bundle matching, it appears that this is not the case for bundle generation, since it is reported that a random 10% of the bundles has been singled out as a test set. Additionally, the statistical significance of the reported results on Tables 3-5 is not assessed.

– (A1-5) We use all user-bundle interactions of the selected 10% bundles as the held-out dataset for bundle generation to evaluate the generation performance on bundles that have not been observed in the training process. As you mentioned, the leave-one-out setting is used only for the bundle matching. We provided a more detailed description of the experimental process in lines 345-346 of the experiments section. In addition, we conduct each experiment at least three times and report the average to secure the statistical significance. We provided the description in line 359 of the experiments section.

• (R1-6) The concluding section is too thin. It should at least report the main points and the lessons learned from the overall approach.

– (A1-6) We learned that each idea in our proposed method helps improve the performance. Especially, the multi-task learning-based approach significantly improves performance because matching and generation are highly related tasks. We provided the lessons learned from the experimental results of our proposed method in lines 448-453 of the conclusion section.

2. Reviewer 2.

• (R2-1) The literature has to be strongly updated with some relevant and recent papers focused on the fields dealt with in the manuscript.

– (A2-1) We added the following related and recent papers to the manuscript.

∗ J. Ma, C. Zhou, P. Cui, H. Yang, and W. Zhu, “Learning disentangled representations for recommendation,” in NeurIPS, 2019.

∗ Q. Deng, K. Wang, M. Zhao, Z. Zou, R. Wu, J. Tao, C. Fan, and L. Chen, “Personalized bundle recommendation in online games,” in CIKM. ACM, 2020.

∗ J. Chang, C. Gao, X. He, D. Jin, and Y. Li, “Bundle recommendation and generation with graph neural networks,” TKDE, 2021.

∗ Z. Wang, Y. Zhu, H. Liu, and C. Wang, “Learning graph-based disentangled representa- tions for next POI recommendation,” in SIGIR. ACM, 2022.

∗ J. Cao, X. Lin, X. Cong, J. Ya, T. Liu, and B. Wang, “Disencdr: Learning disentangled representations for cross-domain recommendation,” in SIGIR. ACM, 2022.

• (R2-2) Making related works as a separate section will be more appropriate for the readers. – (A2-2) We separated the sections of problem definition and related works in lines 56-120.

• (R2-3) Any hyperparamter search done? Authors set default hyperparaters suggested by the articles in reference. Provide explanation regarding this.

– (A2-3) We denoted the search space of hyperparameters and explained that we selected the best among the candidates in lines 360-364 of the experiments section.

• (R2-4) Why recall@k and nDCG@k is used as a performance metric?

– (A2-4) The two metrics are the most widely used metrics for evaluating recommendation accuracy. We added the description for the evaluation metrics in lines 338-339 of the experi- ments section.

---

## [Decision Letter · Decision Letter 1]

5 Jan 2023

Accurate Bundle Matching and Generation via Multitask Learning with Partially Shared Parameters

PONE-D-22-30342R1

Dear Dr. Kang,

We’re pleased to inform you that your manuscript has been judged scientifically suitable for publication and will be formally accepted for publication once it meets all outstanding technical requirements.

Kind regards,

Sathishkumar V E

Academic Editor

PLOS ONE

Additional Editor Comments (optional):

Reviewers' comments:

Reviewer's Responses to Questions

**Comments to the Author**

1. If the authors have adequately addressed your comments raised in a previous round of review and you feel that this manuscript is now acceptable for publication, you may indicate that here to bypass the “Comments to the Author” section, enter your conflict of interest statement in the “Confidential to Editor” section, and submit your "Accept" recommendation.

Reviewer #2: (No Response)

2. Is the manuscript technically sound, and do the data support the conclusions?

Reviewer #2: (No Response)

3. Has the statistical analysis been performed appropriately and rigorously? 

Reviewer #2: (No Response)

4. Have the authors made all data underlying the findings in their manuscript fully available?

Reviewer #2: (No Response)

5. Is the manuscript presented in an intelligible fashion and written in standard English?

Reviewer #2: (No Response)

6. Review Comments to the Author

Reviewer #2: (No Response)

7. PLOS authors have the option to publish the peer review history of their article (what does this mean?). If published, this will include your full peer review and any attached files.

Reviewer #2: **Yes: **Usha Moorthy

---

## [Editor Report · Acceptance letter]

9 Jan 2023

PONE-D-22-30342R1 

Accurate Bundle Matching and Generation via Multitask Learning with Partially Shared Parameters 

Dear Dr. Kang:

I'm pleased to inform you that your manuscript has been deemed suitable for publication in PLOS ONE. Congratulations! Your manuscript is now with our production department. 

Kind regards, 

on behalf of

Dr. Sathishkumar V E 

Academic Editor

PLOS ONE